# Polyisobutylene—New Opportunities for Medical Applications

**DOI:** 10.3390/molecules26175207

**Published:** 2021-08-27

**Authors:** Dóra Barczikai, Judit Domokos, Dóra Szabó, Kristof Molnar, David Juriga, Eniko Krisch, Krisztina S. Nagy, Laszlo Kohidai, Carin A. Helfer, Angela Jedlovszky-Hajdu, Judit E. Puskas

**Affiliations:** 1Laboratory of Nanochemistry, Department of Biophysics and Radiation Biology, Semmelweis University, Nagyvárad tér 4, H-1089 Budapest, Hungary; barczikai.dora@gmail.com (D.B.); Juriga.David@med.semmelweis-univ.hu (D.J.); s.nagykriszti@gmail.com (K.S.N.); 2Institute of Medical Microbiology, Semmelweis University, Nagyvárad tér 4, H-1089 Budapest, Hungary; djudit90@gmail.com (J.D.); szabo.dora@med.semmelweis-univ.hu (D.S.); 3Department of Food, Agricultural and Biological Engineering, College of Food, Agricultural, and Environmental Sciences, The Ohio State University, 222 FABE, 1680 Madison Avenue, Wooster, OH 44691, USA; molnar.182@osu.edu (K.M.); molnarnekrisch.1@osu.edu (E.K.); helfer.12@osu.edu (C.A.H.); 4Department of Genetics, Cell- and Immunobiology, Semmelweis University, Nagyvárad tér 4, H-1089 Budapest, Hungary; kohlasz2@gmail.com

**Keywords:** polyisobutylene, electrospinning, fiber mat, cell adhesion, biofilm formation, COVID mask

## Abstract

This paper presents the results of the first part of testing a novel electrospun fiber mat based on a unique macromolecule: polyisobutylene (PIB). A PIB-based compound containing zinc oxide (ZnO) was electrospun into self-supporting mats of 203.75 and 295.5 g/m^2^ that were investigated using a variety of techniques. The results show that the hydrophobic mats are not cytotoxic, resist fibroblast cell adhesion and biofilm formation and are comfortable and easy to breathe through for use as a mask. The mats show great promise for personal protective equipment and other applications.

## 1. Introduction

Polyisobutylene (PIB) is a unique elastomeric macromolecule, with exceptional impermeability to gases and moisture, high damping and excellent chemical and oxidative stability [1]. Commercial applications of PIB-containing materials include tire inner liners and tubes, sealants, adhesives, condenser caps, pharmaceutical stoppers and chewing gum. The focus of this paper is opportunities for medical applications because PIB possesses excellent tissue and blood compatibility [2]. A linear poly(styrene-*b*-isobutylene-*b*-styrene) triblock copolymer (L_SIBS) (Figure 1) is used in clinical practice as a drug-eluting coating on the TAXUS^TM^ coronary stent [3,4]. Over 10 million patients have benefited from this device, emphasizing the significance of PIB-based biomaterials. Due to the unique low permeability of L_SIBS, sustained drug delivery is achieved, but only approximately 10% of the encapsulated drug, Taxol, is eluted from the coating, which is therapeutic for this application. L-SIBS is commercially available from Kaneka Co., Osaka, Japan. After the success of L_SIBS, other generations of TPE were developed by the Puskas group, such as Arbomatrix©, comprising a branched (arborescent or dendritic) PIB core and end blocks of polystyrene or its derivatives [5,6,7,8], and Allomatrix©, poly(alloocimene-*b*-isobutylene-*b*-alloocimene) [9,10,11,12,13] (Figure 1), but these are not available commercially yet. 

Arbomatrix© and its carbon composite were shown to be bioinert in a rabbit model [13]. ElectroNanospray^TM^, a technology of generating high-velocity spray of nanoparticles, was used to coat several batches of Arbomatrix© polymers loaded with dexamethasone (DXM), a model drug, onto coronary stents. This particulate coating did not have an initial burst release but exhibited slow continuous release over time (20–40% release in 28 days) [14]. Electrospinning is a continuous method for the synthesis of nano- and microfibrous sheets with the aid of a high electric field [15,16]. For biomedical applications requiring more release, it was theorized that encapsulating drugs into electrospun fiber mats would provide a high surface area to volume ratio to release more drug. However, Liu et al. reported that neat L_SIBS could not be electrospun, attributing this to the non-conductivity of the polymer solution [17,18]. Later, we reported conditions under which neat L_SIBS and Arbomatrix© were electrospun onto aluminum stubs and used as a scaffold to grow tissue from chondrocytes [19]. Subsequently, a new method that produced self-supporting fiber mats by electrospinning from mixtures of Arbomatrix© and Allomatrix© and low-molecular-weight poly(ethylene glycol) (PEG) was developed [20,21]. The ratio of TPE to PEG was chosen to be 80/20 *w*/*w*% based on scouting experiments. X-ray photoelectron spectroscopy (XPS) analysis showed that the PEG was completely encapsulated into the fibers. The fiber mats were highly water repellent, with water contact angles (WCAs) > 120°. A model drug, zafirlukast (ZAF), was successfully encapsulated into Arbomatrix© fiber mats and demonstrated greater than 90% release. In the case of physisorbed ZAF, 100% release was achieved in 24 h. Although electrospinning can produce ultrafine fibers, the mean fiber size for ZAF-loaded Arbomatrix©/PEG fibers was larger (4.197 ± 0.580 μm) [20]. However, this system showed higher doses and slower release rates than a study using poly(lactic-co-glycolic acid) polymer coating with a similar drug for reducing the capsular contracture (an inflammatory response around silicone rubber breast prostheses) in vivo [22]. Electrospinning of Allomatrix© has also been successful [23]. Allomatrix© is much easier to synthesize than Arbomatrix© [10,11,12]. The bulk polymer has higher tensile strength (15 MPa and 600% elongation at break) than Arbomatrix© (5.6 MPa and 290% elongation at break). XPS showed that PEG was fully embedded into the electrospun Allomatrix© fibers, similarly to other PIB-based TPE fibers. The tensile strength of the fiber mats measured on microdumbbells was 2.7 MPa at 537% elongation. These rubbery fiber mats were also found to be highly hydrophobic, and cell culture studies showed their cytocompatibility [19]. Based on these favorable properties, these fiber mats showed great promise for tissue scaffold and drug delivery applications.

Arbomatrix© and Allomatrix© have a much higher molecular weight than L-SIBS and can be reinforced with fillers, while SIBS cannot be reinforced [23,24]. However, only SIBS is available commercially. Therefore, we developed a new compound based on SIBS and butyl rubber, with zinc oxide (ZnO) as reinforcing filler. This new compound was successfully electrospun into self-supporting fiber mats. Initial test results of these novel electrospun fiber mats based on this compound will be presented in this article.

## 2. Results and Discussion

### 2.1. Characterization of the Mats

Figure 2a,d display the SEM images of 203.75 g/m^2^ and a 295.5 g/m^2^ mats. It can be seen that the phase morphology of the mats supplied by SNS Nanofiber Technology is more similar to electro-blown materials, with bead-on-string structures and larger particles [25]. More traditional electrospinning yields thinner mats (~10–20 g/m^2^) that are not self-supporting. The WCA was measured to be 134.9° ± 2.4° and 134.2°± 0.4° on the 203.75 and 295.5 g/m^2^ mats respectively (Appendix A).

Therefore, these mats are very hydrophobic materials. XPS analysis at 90° probes ~10 nm from the surface was completed. Figure 3 shows the survey spectra of 203.75 g/m^2^ with peaks representing C, O and Si atoms. Based on the atomic %s (At%) in Table 1, the O and Si signals are minor and likely represent contaminants. In line with earlier studies, no Zn was found on the surface by XPS. One unique property of PIB-based TPEs is very low surface energy, so PIB automatically segregates to the surface, encapsulating particles in the solvent mixture without the need for coaxial spinning [20,21]. Thus, the ZnO particles seem to be visible in Figure 2a,d are coated by a thin (~10 nm) layer of PIB.

### 2.2. Cytotoxicity

Figure 4 presents the test results. According to the statistical analysis, 24 h long treatments of the cells with the release medium in contact with the mats did not alter the cell viability compared to the control (cells only). However, the 72 h treatment reduced the number of viable cells by about 25% compared to the control. Based on these results, the investigated ZnO-containing electrospun mats do not inhibit growth of normal fibroblasts but slow down their proliferation.

The viability data are in line with the morphological observations (Figure 5), which demonstrate that the examined ZnO-containing electrospun mats do not influence the morphology of the fibroblasts.

### 2.3. Cell Adhesion

The polymer and/or the ZnO particles had a moderate green autofluorescence emission, as shown in Figure 6. The results of the two-photon microscopy suggest that fibroblast cells do not adhere to the mats, as the characteristic red color of Vybrant DiD could not be detected either in the case of the 24 h or 72 h incubation. The lack of cell adhesion can be explained with the hydrophobic character of the mats, as well as the unique characteristics of PIB-based molecules. Surface wettability is an important factor that has a strong effect on cell adhesion. Some studies confirmed that there is an optimum value for water contact angle that enhances cell adhesion. Below or above this value, a decrease in adhesion can occur. [26]. PIB-based TPEs were shown to have low protein adsorption of 256 ng/cm^2^ compared to 590 ng/cm^2^ for silicone rubber [27,28].

### 2.4. Antibacterial Activity

ZnO is used for various applications, including as a reinforcing white filler for rubber [29] or as an antimicrobial agent [30]. ZnO nanoparticles were shown to be effective for this latter application. Thus, the fiber mats were tested for antimicrobial activity. Appendix A shows the arrangement of the Kirby–Bauer test used for the evaluation of the antibacterial activity of the samples. In the first test, bacteria were incubated in a liquid culture medium together with the ZnO-containing mats. After 24 h, optical density (OD) was measured and compared to that of the bacteria suspension without mats (control). The OD did not change upon contact with the mats. This is in line with the finding that the ZnO is completely coated with PIB, and very little, if any, is released into the liquid. The standard Kirby–Bauer test showed the same results, revealing no antibacterial activity.

After 24 h of incubation at 37 °C, no diffusion zone (antibacterial agent inhibited bacterial growth) could be observed in any cases (Figure 7a and Appendix A). To evaluate the area under the samples, meshes and paper discs were removed. No bacterial growth could be observed under the mats (Figure 7b and Appendix A) or the paper discs, indicating that the lack of bacterial growth under the specimens is not due to the antibacterial effect of the ZnO-containing samples.

Thus, no antibacterial activity against *E. coli* or *S. epidermidis* was found with the mats. This can be attributed to the fact that according to XPS, there were no ZnO particles on the surface of the fibers, and very little if any ZnO diffused out of the mats. 

### 2.5. Biofilm Formation

According to previous research articles [30], *P. aeruginosa* can only form a biofilm when glucose is present in the medium. In the presence of glucose, a robust biofilm can be observed with optical density (OD) values typically between 0.2 and 0.4 and as high as 0.5 if salt is added to the medium [31]. Upon inhibition, OD at 595 nm typically reduces below 0.1. The OD data presented in Figure 8 demonstrate that the investigated mats mitigated the biofilm-forming capacity of *P. aeruginosa*

### 2.6. Potential Applications

One potential application of fiber mats is wound healing. The lack of cell adhesion is quite a favorable characteristic in the field of wound care. Matrices that exhibit anti-adhesive properties can usually be easily removed even from serious injuries and skin ulcers, in contrast to traditional approaches of wound care [32]. This feature, together with the lack of cytotoxicity, makes fiber mats quite promising for potential wound care application.

Another potential application is for use in personal protective equipment (PPE) [33,34]. Specifically, the use of fiber mats for face masks to protect from COVID-19 is of current great interest. The virus is 62 ± 8 nm in size based on atomic force microscopy [35] and often travels in biological aerosols from coughing or sneezing that are 0.5–3 microns in size and prefer hydrophilic surfaces [36]. Current N95 masks have high filtration efficiency but are thicker than surgical masks and rather uncomfortable for long-term use in daily life. N95 masks also impede breathing, and users experience problems due to the increase in temperature and humidity between the face and the mask. Face masks containing a thin layer of electrospun poly(vinylidene fluoride) (PVDF), polycaprolactone (PCL) or nylon fiber mats (<10 g/square meter) are currently marketed [33]. The fiber mat filters are claimed to have a 99.9% barrier efficiency against viruses, but the mats are not self-supporting, so they must be incorporated into a multi-layer structure. The mats discussed in this paper are self-supporting at ~2–300 g/m^2^. They can be sterilized by several methods, including ethylene oxide, low-pressure plasma or UV treatment, or chlorine dioxide, and can be recycled by simply dissolving them in a solvent and re-spinning into a new mat. These stretchable mats could be attached to rubber frames, creating a tight fit around the nose and mouth, to provide an effective, more comfortable, face mask. Filtering efficiency tests specify corresponding pressure drop requirements, which are presented in Table 2.

Preliminary testing of five experimental fiber mats with different thicknesses (13.8–42.6 g/m^2^) showed that filtering efficiency is directly correlated to the pressure drop across the mats, as shown in Figure 9a. At 100% efficiency, a 250 Pa pressure drop is expected, below the maximum specified by US and Chinese standards and close to the EU standard. It is very easy to breathe through a mask fashioned from the mats. An example is shown in Figure 9b.

Detailed filter efficiency studies are in progress.

## 3. Materials and Methods

### 3.1. Materials

The proprietary compound contained SIBS, butyl rubber and ZnO and was mixed by Hexpol (Burton, OH, USA). Semi-commercial electrospun fiber mats were supplied by SNS Nanofiber Technology (Hudson, OH, USA) using their proprietary method, so the conditions used were not revealed. Dimethyl sulfoxide (anhydrous, ≥99.94%, Reanal, Budapest, Hungary), ultrapure water (Water Purification System, Zaneer, Human Corporation, Seoul, Korea), phosphate-buffered saline (Sigma Aldrich, Darmstadt, Germany), natrium-azid (Sigma Aldrich, Darmstadt, Germany), normal human dermal fibroblasts (NHDFs, PromoCell GmbH, Heidelberg, Germany), fibroblast growth medium (ready-to-use) (PromoCell GmbH, Heidelberg, Germany), SupplementMix fibroblast growth medium (PromoCell GmbH, Heidelberg, Germany), trypsin (Gibco, Waltham, MA, USA), paraformaldehyde (4% Sigma Aldrich, Darmstadt, Germany), Hoechst 33342 (Bisbenzimide, Thermo Fisher Scientific, Waltham, MA, USA) Vybrant™ DiD Cell-Labeling Solution (Thermo Fisher Scientific, Waltham, MA, USA), CellTiter-Glo^®^ Luminescent Cell Viability Reagent (Promega, Madison, WI, USA), chlorine dioxide (3350 ppm, Solvocid, Budapest, Hungary), Mueller–Hinton agar (Biolab Zrt., Budapest, Hungary), *E. coli* ATCC 25922, *S. epidermidis* ATCC 14990, *P. aeruginosa* ATCC 27853, Brain Heart Infusion (BHI) broth (Mast Group Ltd., Merseyside, UK), Syringe Filter ISO9001:2008, 0.22 μm filter mats (Thermo Fisher Scientific, Waltham, MA, USA), Nunclon Sphera 24-Well Plate (Thermo Fisher Scientific, Waltham, MA, USA), 96-well white/clear-bottomed plate (Thermo Fisher Scientific, Waltham, MA, USA).

### 3.2. Scanning Electron Microscopy (SEM)

The microstructure of the mats was examined with scanning electron microscopy utilizing a secondary electron detector. For SEM analysis, samples were placed on conductive carbon tape and sputter-coated with a 20–30 nm layer of gold using a sputter coating system (model: JFC-1200, JEOL Ltd., Tokyo, Japan). Images were taken with a scanning electron microscope (model: JSM 6380LA, JEOL Ltd., Tokyo, Japan). The applied voltage was 10 kV and micrographs were obtained at 1500×, 500× and 100× magnifications. 

### 3.3. X-ray Photoelectron Spectroscopy (XPS)

Measurements were carried out using a Kratos Axis Ultra XPS (Manchester, UK) with a monochromic Al Kα source (1486.6 eV, 12 kV, 10 mA) and a charge neutralizer. Survey spectra were collected with 100 eV pass energy, while high-resolution spectra were collected with 20 eV pass energy. The CasaXPS software was used for data analysis. The corresponding reference signal was the C1s signal with a binding energy of 285 eV. Curve fitting was performed using the Gaussian–Lorentzian distribution with the deduction of the Shirley background.

### 3.4. Water Contact Angle (WCA) Measurement

Water contact angle (WCA) was evaluated for both compositions using an OCA 15 Plus instrument with a built-in camera (Dataphysics, Filderstadt, Germany). Small circular samples were cut from the mats, and a droplet of distilled water (5 μL) was placed onto the samples from a 50 μL Hamilton syringe equipped with a 0.56 mm inner diameter needle. 

### 3.5. Cytotoxicity Assay

NHDF normal human dermal fibroblasts (PromoCell) were cultivated in the fibroblast growth medium with 2% supplement provided by the manufacturer (PromoCell). On Day 1, cells were seeded into two white-walled cell culture plates with 96 wells each to reach 5000 cell/cm^2^ (1850 cells/well in 100 µL of culture medium/well). Discs of 12 mm diameter were cut from the mats and were sterilized in 1% ClO_2_/PBS solution for 30 min, after which they were dried under a sterile hood [38]. Subsequently, the sterile discs were immersed individually into 1 mL of cell culture medium and stored at 37 °C for 24 h. After 24 h, the old growth medium on the cells was replaced by the release medium of the mats in each well except the control wells. The cells were incubated further at 37 °C. For one of the cell culture plates, the incubation time was 24 h, while for the other plate, it was 72 h. After the incubation period, the cells were treated with CellTiter-Glo reagent according to the instructions provided by the manufacturer, and cell viability was measured based on detecting luminescence with a Fluoroskan™ FL Microplate Fluorometer (Thermo Fisher Scientific, Waltham, MA, USA) and luminometer. The measured luminescence values of the cells treated with the release medium of the fiber mats were compared to luminescence of the cells treated with normal fibroblast growth medium (control). Additionally, a morphological study utilizing a Zeiss Axio Observer A1 inverted microscope (Carl Zeiss AG, Oberkochen, Germany) was performed both after 24 h and 72 h incubations as well. A non-parametric one-way ANOVA analysis (Kruskal–Wallis test) was carried out on cell viability data in order to decide whether the differences between the samples are significant or not (*p* < 0.05). Statistical evaluation was performed using the STATISTICA 10 software.

### 3.6. Cell Adhesion

To evaluate the adherence of fibroblast to the mats, a direct cell-seeding method was performed. Two discs with a diameter of 12 mm were prepared from each mat (one for the 24 h and one for the 72 h test). They were sterilized in chlorine dioxide solution (diluted 100 times) for 30 min and dried under a sterile hood [38]. The disks were placed into a Nunclon Sphera 24-Well Plate (with super low cell attachment surface) and glass rings were placed onto the top of the disks to hold them in place. Fibroblasts were stained with vital dye Vybrant DiD according to the protocol provided by the manufacturer. Then, stained cells were seeded onto the surface of the membranes (20,000 cells/well, 27,000 cell/cm^2^, 500 μL medium/well). This required 50 μL of cell suspension and 450 μL of clear cell culture medium to reach the final volume of 500 µL. The 450 µL of culture media was evenly distributed between the inner and outer parts of the glass rings after incubating the samples with 50 μL of cell suspension for 30 min at 37 °C. After 24 h, cells were fixed by adding 4% of paraformaldehyde (PFA). After 20 min, PFA was removed and nuclear staining was carried out by adding Hoechst 33342 (bisbenzimide). The samples were kept at 4 °C away from any source of light until further examination. After 72 h, the same fixing procedure was performed on the 72 h samples. Samples were removed from PBS right before the examination. A Femto2D series (Femtonics, Budapest, Hungary) multiphoton microscope was used for the measurement. Photoactivity of samples was induced with a DeepSee^TM^ laser (Spectra Physics, Milpitas, CA, USA). The resolution of the images is 1.33 µm/pixel.

### 3.7. Antibacterial Testing

Antibacterial activity was evaluated using two methods: a liquid-phase pre-screening and the standardized Kirby–Bauer disc diffusion method. Two bacterial strains were used, namely *S. epidermidis* and *E. coli*. Prior to the antibacterial evaluation, membrane discs with a diameter of 6 mm were cut from the mats (n = 8 per sample and per bacteria strain), and the mass of each disc was measured and recorded. Discs were sterilized using chlorine dioxide in 100-fold dilution with phosphate-buffered saline (PBS, pH = 7.4, I = 0.15 M). Discs were immersed into 2 mL of chlorine dioxide solution, and they were removed after 30 min and then washed with pure PBS to remove chlorine dioxide residues. In the first method, bacteria were incubated in a liquid culture medium together with the ZnO-containing mats. After 24 h, optical density (OD) was measured and compared to that of bacteria suspension without mats (control). In the standard Kirby–Bauer test, bacteria colonies from each strain were immersed individually into 5 mL of saline, and then the suspensions were diluted to reach a 0.5 McFarland turbidity using a DEN-1 Benchtop Densitomer (Grant Instruments, Shepreth, UK). A sterile swab was then immersed into the bacteria-containing saline, after which the Muller–Hinton agar was covered with the bacteria-containing solution using the swab. These sterile disks were placed onto the top of the inoculated Muller–Hinton agar and then the plates were incubated for 24 h at 37 °C. After 24 h, the diameter of the bacteria-free zone (diffusion zone) was measured for each disc. Because the mats have a highly hydrophobic characteristic, in the case of four discs (out of 8 per mat and per bacteria strain), 5 μL of dimethyl sulfoxide (DMSO) was dribbled on the surface of the chosen discs to enhance the wetting of the mats. DMSO is often used in microbiological evaluation to promote the release of drug with poor water solubility. For negative control, pure paper discs were used (n = 2 per sample and per bacteria), and on one of the two paper discs, a drop of DMSO was also dribbled.

### 3.8. Biofilm Formation Testing

The method was performed according to the following procedure [39]: *P. aeruginosa* was inoculated in a 3–5 mL culture and grown to stationary phase (overnight culture) under aerobic conditions. A 500 μL volume of the overnight culture was mixed with 50 mL of 2 m/m% glucose/BHI (brain heart infusion) culture medium (previously filtered using a Syringe Filter ISO9001:2008, 0.22 μm filter membranes) [39]. Discs (diameter = 13 mm) were sterilized as described in Section 2.6. Sterile discs were placed into the separate wells of a 24-well plate (Nunclon Sphera 24-Well Plate, Thermo Fisher Scientific, Waltham, MA, USA) and 1 mL of diluted bacteria culture was pipetted into each well. The 24-well plate was then incubated for 72 h at 37 °C. The mats were removed from the bacterium culture and placed into a clean 24-well plate and thoroughly washed with PBS to remove colonies that were not directly attached to the samples. A 500 μL volume of 0.1 *w*/*V*% crystal violet/water was added to the samples and samples were stained for 10 min at room temperature. Additional control samples (discs without bacterial biofilm) were also stained with crystal violet at this point. Crystal violet solution was removed, and samples were washed with PBS four times to remove any excess dye. A 1 mL volume of 30 *V*/*V*% acetic acid was added to the samples to solubilize crystal violet. Samples were incubated for 15 min at room temperature. 

Finally, 100 μL of crystal violet/acetic acid solution was transferred to a clean 96-well plate, and optical density (OD) values were measured at two wavelengths between 500 and 600 nm (560 and 595 nm, typical measurement wavelengths for crystal violet) (SkanIt Software 5.0 for Microplate Readers RE, ver. 5.0.0.42). OD values of the control samples were subtracted from those of the treated samples.

### 3.9. Filtration Efficiency Tests

These tests were carried out according to the U.S. NIOSH (National Institute for Occupational Safety and Health) N95 filtering facepiece respirator (FFR) certification method. Under these test conditions, NIOSH specifies a maximum inhalation resistance of 0.34 kPa and maximum exhalation resistance of 11 kPa [37].

## 4. Conclusions

Self-supporting rubbery fiber mats that were electrospun from a ZnO-containing proprietary polyisobutylene-based compound were found to be very hydrophobic. The mats were not cytotoxic, resisted fibroblast cell attachment and mitigated the biofilm-forming capacity of *P. aeruginosa.* However, they did not exhibit antimicrobial properties, which is due to the low permeability of polyisobutylene covering the ZnO particles. These self-supporting fiber mats are very promising materials for applications such as PPE or wound healing.

## Figures and Tables

**Figure 1 molecules-26-05207-f001:**
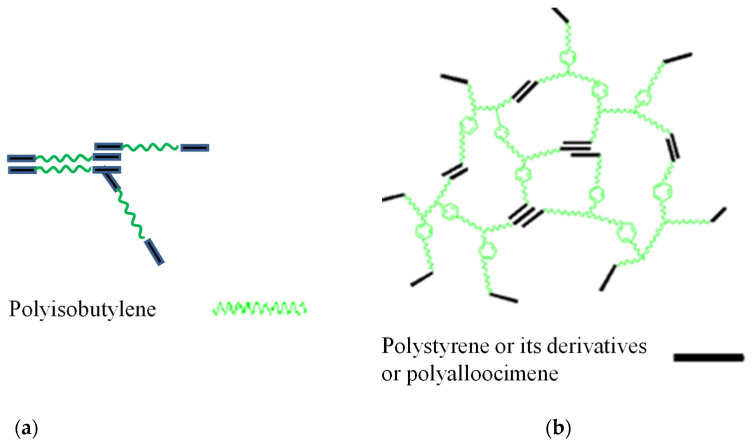
Sketch of the architecture and phase separation of L-SIBS and Allomatrix© (**a**) and Arbomatrix© (**b**).

**Figure 2 molecules-26-05207-f002:**
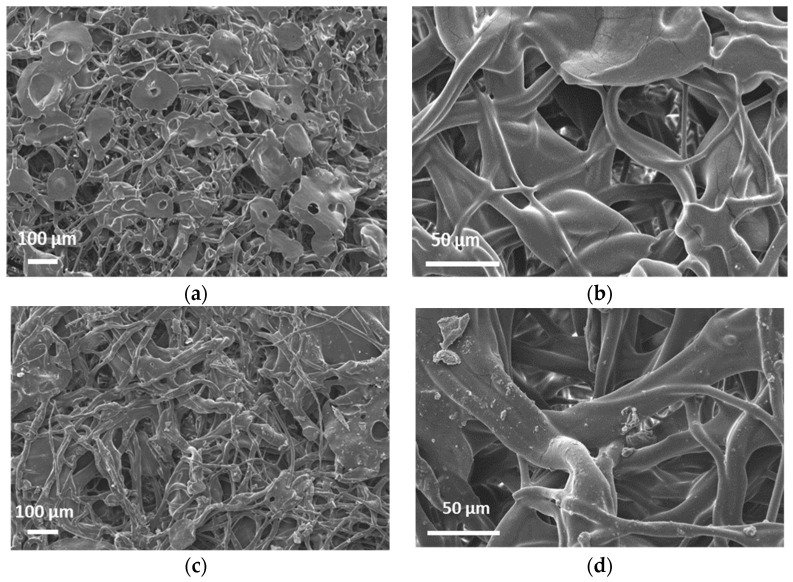
SEM micrographs of the 203.75 g/m^2^ (**a**,**b**) and the 295.5 g/m^2^ mats (**c**,**d**).

**Figure 3 molecules-26-05207-f003:**
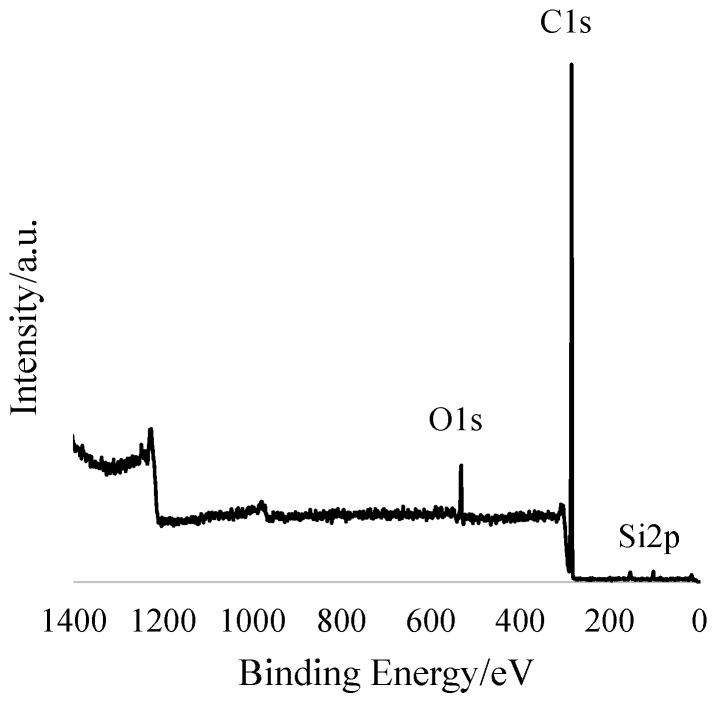
XPS survey spectrum of the 203.75 g/m^2^ sample.

**Figure 4 molecules-26-05207-f004:**
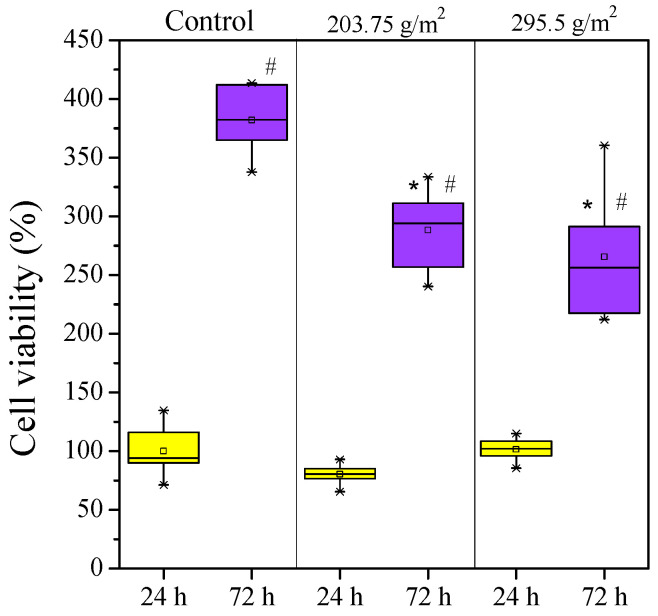
Box plot and statistical analysis of cell viability study of NHDF cells in the presence of SIBS/ZnO matrices. * *p* < 0.05 compared to the control at the same time point, ^#^ *p* < 0.05 compared to the previous time point in the same group.

**Figure 5 molecules-26-05207-f005:**
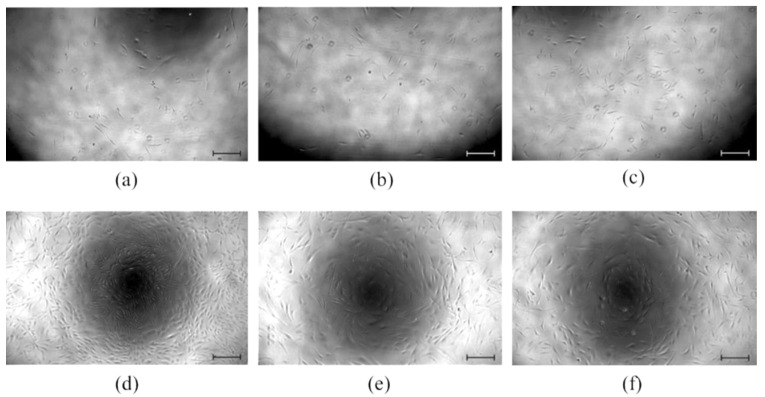
Morphological study of NHDF cells after 24 h in the presence of (**a**) pure cell medium (control) and release medium of (**b**) 203.75 g/m^2^ sample, (**c**) 295.5 g/m^2^ sample and after 72 h in the presence of (**d**) pure cell medium (control) and release medium of (**e**) 203.75 g/m^2^ sample and (**f**) 295.5 g/m^2^ sample (scale bar is 200 μm).

**Figure 6 molecules-26-05207-f006:**
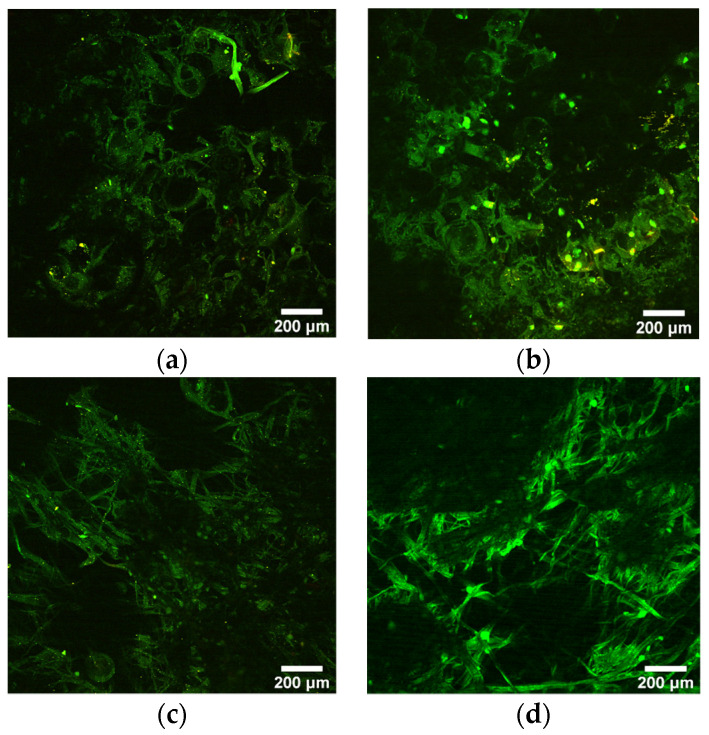
Multiphoton images of 203.75 g/m^2^ mat at 24 h (**a**) and 72 h (**b**) and 295.5 g/m^2^ mats at 24 h (**c**) and 72 h (**d**) after cell seeding. Samples show low autofluorescence in green.

**Figure 7 molecules-26-05207-f007:**
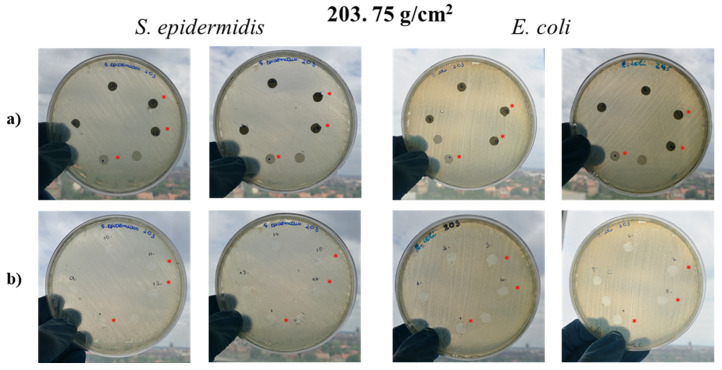
Petri dishes after the first 24 h incubation (**a**) and Petri dishes after removal of specimen discs (**b**). * indicates samples treated with DMSO.

**Figure 8 molecules-26-05207-f008:**
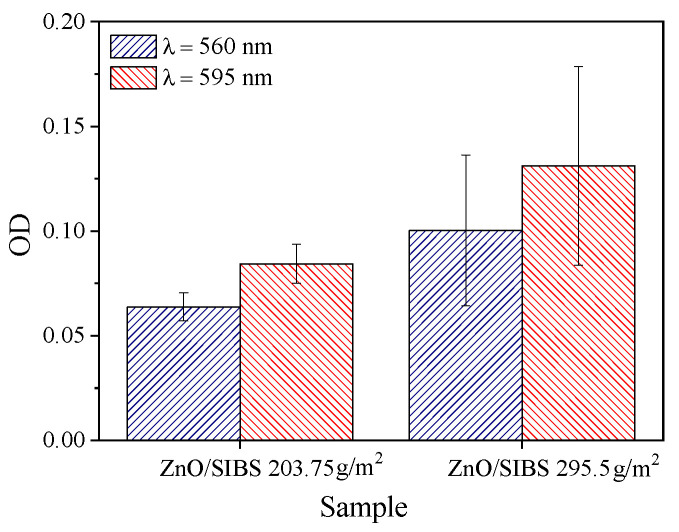
OD values of *P. aeruginosa* biofilms on the fiber mats.

**Figure 9 molecules-26-05207-f009:**
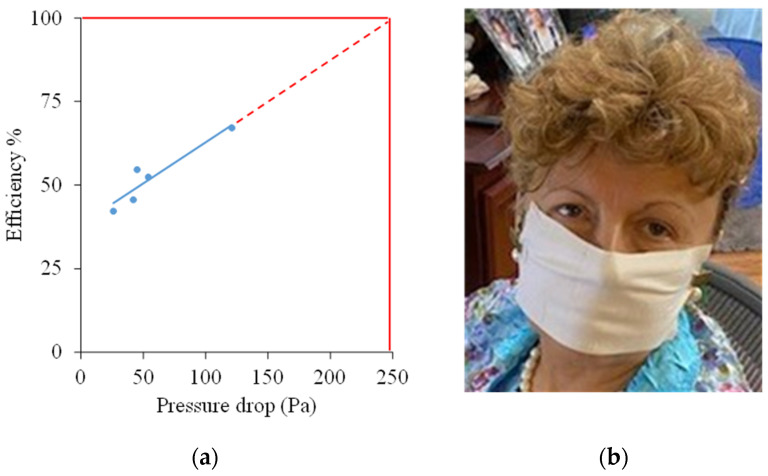
Correlation of filtering efficiency and pressure drop across the mats (**a**); potential COVID-19 mask (**b**).

**Table 1 molecules-26-05207-t001:** Composition of 203.75 g/m^2^ and 295.5 g/m^2^ mats based on XPS survey spectra.

Atoms	Binding Energy (eV)	203.75 g/m^2^ At%	295.5 g/m^2^ At%
**O**	532	4.47	5.23
**C**	285	93.38	93.00
**Si**	102	2.15	1.77

**Table 2 molecules-26-05207-t002:** Maximum allowed pressure drop for PPEs and surgical/medical masks per different standards around the globe [37].

	N95, FFP2, KN95		Surgical/Medical Mask Material	Cloth Mask Material
	Inhalation	Exhalation		
**NIOSH**	343 Pa (85 L/min)	245 Pa (85 L/min)	---	---
**ASTM**	---	---	50/60/60 Pa/cm^2^ (245/294/294 Pa) (8 L/min)	---
**EU**	240 Pa (95 L/min) 70 Pa (30 L/min)	300 Pa (160 L/min)	40/40/60 Pa/cm^2^ (196/196/294 Pa) (8 L/min)	70 Pa/cm^2^
**China**	350 Pa (85 L/min)	250 (85 L/min)	49 Pa/cm^2^ (240 Pa) (8 L/min)	---

## Data Availability

The data presented in this study are available on request from the corresponding author.

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
