# Peer review of "Polyisobutylene—New Opportunities for Medical Applications"

_molecules, 2021, doi:10.3390/molecules26175207_

Round 1
Reviewer 1 Report
A collaborative paper from the Hungarian Semmelweis University and the Ohio State University (US) describes a new polyisobutylene-based hydrophobic mat. This material is mentioned to be useful in wound healing or as a protective mask. The paper is well written, and I highly support its publication in Molecules, but only after the Authors answered my comments, especially the last ones.
1) Maybe the experimental part (3. Materials and Methods) should be presented before the 2. Results and Discussion part. The reader can better understand the first sentences of this, i.e., the results part.
2) The part “2.4 Antibacterial activity“ does not provide any positive results but contains three figures. Two figures should be moved to supplementary materials.
3) There is no Figure 10; Figure 11 comes right after Figure 9.
4) „The proprietary compound contained SIBS, butyl rubber and ZnO and was mixed by Hexpol (Burton, OH, USA).”
It would be great to know a bit more about the material.
“Semicommercial electrospun fiber mats were supplied by SNS Nanofiber Technology (Hudson, OH, USA) using their proprietary method so the conditions used were not revealed.”
In such research papers, reproducibility is a crucial aspect. If the method or the material is not yet patented, the authors maybe have to wait until it is, but they definitely have to provide some info about the material they work with in this publication.
Author Response
Please, see the attachment.

Reviewer 2 Report
In this work, the author designed and fabricated PIB-based compound containing zinc oxide (ZnO) was electrospun into self-supporting mats with excellent hydrophobic property, non-cytotoxic, resist fibroblast cell adhesion and biofilm formation. The outstanding self-supporting rubbery fiber mats are promising materials for wound healing and mask. This work may be interesting for biomaterials field. However, I have some doubt in this work about the materials. So, the following issues the authors should address before this paper can be considered for publication in molecules.
- The third paragraph of the introduction is repeated with the second paragraph, please revise.
- In this work, the self-supporting mats with ZnO as reinforcing filler. So, the author should provide corresponding dates for the influence of ZnO content on the self-supporting mats.
- How to control the self-supporting mats is 203.75g/m2 or 295.5g/m2, please provide a detailed explanation.
- The author should provide the detailed fabrication process of the PIB-based compound containing zinc oxide (ZnO) self-supporting mats.
- More references should be cited in this work, such as Journal of Energy Chemistry, 2021, 58, 397-407; Polymers 2019, 11, 1718; Journal of Hazardous Materials, 2020, 395, 122639
Author Response
Please, see the attachment.

Round 2
Reviewer 1 Report
I suggest: accept in present form